# Separation and Enrichment of Three Coumarins from *Angelicae Pubescentis* Radix by Macroporous Resin with Preparative HPLC and Evaluation of Their Anti-Inflammatory Activity

**DOI:** 10.3390/molecules24142664

**Published:** 2019-07-23

**Authors:** Yuqiao Yang, Ruichao Zhu, Jin Li, Xuejing Yang, Jun He, Hui Wang, Yanxu Chang

**Affiliations:** 1Tianjin State Key Laboratory of Modern Chinese Medicine, Tianjin University of Traditional Chinese Medicine, Tianjin 300193, China; 2Tianjin Key Laboratory of Phytochemistry and Pharmaceutical Analysis, Tianjin University of Traditional Chinese Medicine, Tianjin 300193, China

**Keywords:** coumarins, macroporous resins, Angelicae Pubescentis Radix, preparative HPLC, anti-inflammatory

## Abstract

In order to enrich and separate three coumarins (columbianetin acetate, osthole and columbianadin) from Angelicae Pubescentis Radix (APR), an efficient method was established by combining macroporous resins (MARs) with preparative high-performance liquid chromatography (PHPLC). Five different macroporous resins (D101, AB-8, DA-201, HP-20 and GDX-201) were used to assess the adsorption and desorption characteristics of three coumarins. The result demonstrated that HP-20 resin possessed the best adsorption and desorption capacities for these three coumarins. Moreover, the adsorption dynamics profiles of three coumarins were well fitted to the pseudo second order equation (R^2^ > 0.99) for the HP-20 resin. The adsorption process was described by the three isotherms models including Langmuir (R^2^ > 0.98, 0.046 ≤ R_L_ ≤ 0.103), Freundlich (R^2^ > 0.99, 0.2748 ≤ 1/n ≤ 0.3103) and Dubinin Radushkevich (R^2^ > 0.97). The contents of columbianetin acetate, osthole and columbianadin in the product were increased 10.69-fold, 19.98-fold and 19.68-fold after enrichment, respectively. Three coumarins were further purified by PHPLC and the purities of them reached above 98%. Additionally, the anti-inflammatory effects of these three coumarins were assessed by Lipopolysaccharide (LPS)-induced RAW 264.7 cells. It was found that the production of NO and MCP-1 was obviously inhibited by three coumarins. Columbianetin acetate, osthole and columbianadin could be used as potentially natural anti-inflammatory ingredients in pharmaceutical products. It was concluded that the new method combining MARs with PHPLC was efficient and economical for enlarging scale separation and enrichment of columbianetin acetate, osthole and columbianadin with anti-inflammatory effect from the APR extract.

## 1. Introduction

*Angelicae Pubescentis* Radix (APR) derived from the roots of *Angelica pubescens* Maxim.f. biserrata Shan et Yuan has been recorded in Chinese pharmacopoeia and is usually used to treat rheumatic arthralgia [1]. The phytochemical studies have demonstrated that approximately 70 coumarins including columbianetin, columbianadin (CBD), osthole (OE), columbianetin acetate (CBA), isoimperatorin, bergapten and columbianetin-β-d-glucopyranoside have been isolated from APR [2]. Our previous study showed that CBA, OE and CBD were the three main coumarin constituents from APR [3]. Their chemical structures are shown in Figure 1. OE and CBD are often selected as markers for the quality control of APR. Pharmacological studies have shown that these three coumarins have anti-inflammatory, analgesic, antidiabetic, anthelmintic and anticancer activities [4,5,6,7,8,9]. Therefore, it is necessary to build a simple, low-cost and efficient purification method for these three coumarins from APR extract, in view of these advantages.

The methods of silica gel column, liquid-liquid extraction and high-speed counter current chromatography have been applied for the separation of coumarins from traditional Chinese medicines [10,11,12]. Unfortunately, these extraction and separation methods are unsuitable for large-scale industrial production due to several drawbacks, including inefficiency, high cost, and poor purity. Recently, macroporous resins (MARs) have been extensively used to separate and purify pharmacologically-active natural products, including flavonoids, phenolic, glycosides, saponins due to high adsorption capacity, environmentally-friendly and easy regeneration properties [13,14,15,16,17,18,19]. However, it has not been reported that three major coumarin compounds were enriched and separated by macroporous resins from APR extract.

The purpose of the current study was to develop a quick and efficient method for separation and enrichment of CBA, OE and CBD from APR extract. In this study, three coumarins (CBA, OE and CBD) of APR extract were separated and enriched with MARs (HP-20) and were further purified by preparative high-performance liquid chromatography (PHPLC). Moreover, it is the first report for the systematical investigation of adsorption/desorption behaviors, kinetics, isotherms and thermodynamics for static adsorption of APR coumarins on macroporous resins. In addition, the anti-inflammatory effects of these three coumarins of APR were also assessed in RAW264.7 cells.

## 2. Result and Discussion

### 2.1. Optimization of Resin

The adsorption, desorption capacity and desorption ratio of three coumarins on resins are listed in Figure 2. Among the five resins (The physical characteristics are listed in Table 1.), D101 (8.82 mg/g), AB-8 (8.82 mg/g) and HP-20 (8.79 mg/g) had higher adsorption capacity than other adsorption resins, and the desorption ratio of the three adsorption resins were 85.72%, 90.84% and 94.76%, respectively. This may be due to their large specific surface area. Although DA-201 and GDX-201 resins exhibited higher desorption ratio, their adsorption capacity were lower than other resins. Taking adsorption capacity and the desorption ratio into consideration, HP-20 resin was used to enrich and purify these three coumarins.

### 2.2. Adsorption Dynamics

Adsorption dynamics describe the adsorption rate of the adsorbate on an adsorbent and the adsorption time from the starting time to equilibrium. Adsorption dynamics curves of three coumarins at 298 K are shown in Figure 3. The adsorption capacity of three coumarins increased quickly in the first 40 min, and then it increased tardily, and reached adsorption equilibrium after approximately 180 min. The adsorption kinetics parameters of three models were presented in Table 2. The pseudo second order model could illustrate the adsorption process of the three coumarins on HP-20 resin better than two other models, because the regression coefficient (R^2^) was nearer to 1 (0.9971, 0.9927 and 0.9942 for three coumarins, respectively). Although the whole adsorption process of the three coumarins could not be adequately described by the intra particle diffusion dynamics model in this study (0.5843 ≤ R^2^ ≤ 0.6952), the process could be divided into 3 steps (Figure 3D), where there was a matching linear relationship that did not go through the origin. This demonstrated that the adsorption processes may be influenced by multi diffusion stages, including the first stage of boundary layer diffusion, the second stage of intra-particle diffusion and the third stage of adsorption equilibrium process.

### 2.3. Adsorption Isotherms and Thermodynamics

Adsorption isotherms were used to explain the adsorption capacity of the absorbate on an absorbent at the different temperature. The adsorption isotherms of the three coumarins on HP-20 resin at 298 K, 308 K and 318 K are shown in Figure 4. With the three coumarins concentration increasing, the equilibrium adsorption capacities increased. Interestingly, the equilibrium adsorption capacities of three coumarins on the HP-20 resin increased with increased temperature, which indicate that the adsorption process was an endothermic. The Langmuir, Freundlich and Dubinin Radushkevich (DR) isotherms parameters and regression coefficient (R^2^) in Table 3 show that the Q_0_ and K_F_ values increased with the temperature, which further demonstrated that high temperature had a positive effect on the equilibrium adsorption capacity. Within the experimental concentration range, the values of R_L_ ranged from 0.046 to 0.103, demonstrating that the adsorption of three coumarins was a preferential [20]. The 1/n values in the Freundlich isotherms model were between 0.2748 and 0.3103, which suggested that the three coumarins could be easily absorbed by the HP-20 resin. All of the adsorption energy E values were higher than 16 kJ/mol. It is confirmed that the adsorption of three coumarins is driven by a chemical process [21].

The thermodynamics equation parameters for three coumarins on HP-20 resin at different temperature are shown in Table 4. The negative ∆G^0^ and positive ∆H^0^ values showed that adsorption process was spontaneous, endothermic. The positive ∆S^0^ illustrated that the random degree at the solid–liquid interface increased.

### 2.4. Dynamic Adsorption and Desorption

The dynamic leakage curve was investigated for getting the appropriate loading volume of APR extracts. As shown in Figure 5A, these three coumarins were absorbed completely by the HP-20 resin before 5 BV. When the loading volume was 5.5 BV, the leakage point was observed. Hence, 5 BV was considered as the appropriate loading volume. As shown in Figure 5B, the desorption ratio was the highest when the ethanol concentration reached 80%. Furthermore, 60% and 80% ethanol could desorb the great majority of three coumarins absorbed by HP-20 resin. Although the 60% ethanol eluted three coumarins, it also eluted other impurities ingredients. Therefore, ultra-pure water (5 BV) and 60% ethanol (5 BV) were firstly used to eliminate impurities. Ethanol (5 BV, 80%) was considered as the best desorb solvent to elute these three coumarins. Comparing the chromatograms of pre-processing and those of post-processing, it was clear that much interference had been eliminated (Figure 6A,B).

### 2.5. Enrichment of Three Coumarins

The contents of CBA, OE and CBD were increased from 0.27%, 1.15%, 0.36% to 2.92%, 22.98%, and 7.16% with a recovery yield of 31.04%, 58.05% and 57.17% by an experiment of lab-scale enrichment, respectively. These results indicated that HP-20 resin could be applied for large scale enrichment of three coumarins in APR.

### 2.6. Separation of Three Coumarins by PHPLC

The PHPLC method was used and optimized to separate and purify these coumarins from the extract. Finally, 280 mg of CBA, 2268 mg of OE and 615 mg of CBD were obtained from 11.77 g of 80% ethanol fraction. The purity of three coumarins was over 98% (Figure 6C–E).

The chemical structures of the purified coumarins were illuminated by the spectral data of MS, ^1^H-NMR and ^13^C-NMR. The detailed data were as follows.

Compound **1**. ESI-MS *m/z*: 288 [M]^+^, 228, 213, 203, 187, 176, 159, 131, 77, 59; ^1^H-NMR (600 MHz, CDCl_3_) *δ*: 6.21 (1H, d, *J* = 9.5Hz, H-3), 7.64 (1H, d, *J* = 9.5Hz, H-4), 7.27 (1H, d, *J* = 8.1Hz, H-5), 6.75 (1H, d, *J* = 8.2Hz, H-6), 3.29 (2H, m, *J* = 7.7, 7.8Hz, H-3′), 5.16 (1H, dd, *J* = 7.6, 7.9Hz, H-2′),1.58 (3H, s, H-4′), 1.52 (3H, s, H-5′), 1.99 (3H, s, OAc-CH3); ^13^C-NMR (150 MHz, CDCl_3_) *δ*:170.2 (OAc-C=O), 160.9 (C-2), 112.2 (C-3), 143.9 (C-4), 128.9 (C-5), 106.7 (C-6), 163.9 (C-7), 113.4 (C-8), 151.2 (C-9), 113.1 (C-10), 27.6 (C-1′), 88.7 (C-2′), 82.1 (C-3′), 22.2 (C-4′), 21.9 (C-5′), 20.9 (OAc-CH3). Comparing the above data with previous study [22], it was identified as columbianetin acetate.

Compound **2**. ESI-MS *m/z*: 244 [M]^+^, 229, 213, 201, 159, 131, 103, 89, 77, 63, 51; ^1^H-NMR (600 MHz, CDCl_3_) *δ*: 6.23 (1H, d, *J* = 8.2Hz, H-3), 7.28 (1H, d, *J* = 8.0 Hz, H-5), 7.61 (1H, d, *J* = 8.0 Hz, H-4), 6.23 (1H, d, *J* = 6.2 Hz, H-3), 7.62 (1H, d, *J* = 9.4 Hz, H-4), 7.29 (1H, d, *J* = 8.6 Hz, H-5), 6.84 (1H, d, *J* = 8.5Hz, H-6), 3.53 (2H, d, *J* = 7.4 Hz, H-1′′), 5.22 (1H, m, H-2′), 1.84 (3H, s, H-4′), 1.67 (3H, s, H-5′), 3.92(3H, s, OCH3); ^13^C-NMR (150 MHz, CDCl_3_) *δ*:160.2 (C-2), 113.0 (C-3), 143.9 (C-4), 126.2 (C-5), 107.4 (C-6), 161.5 (C-7), 112.9 (C-8), 152.8 (C-9), 117.9 (C-10), 21.9 (C-1′), 121.1 (C-2′), 132.7 (C-3′), 25.8 (C-4′), 17.9 (C-5′), 56.1 (OCH3). Comparing the above data with previous study [23], it was identified as osthole.

Compound **3**. ESI-MS *m/z*:328 [M]^+^, 228, 213, 187,159, 131, 83, 55, 43; ^1^H-NMR (600 MHz, CDCl_3_) *δ*:6.21 (1H, d, *J* = 9.5Hz, H-3), 7.64 (1H, d, *J* = 9.5Hz, H-4), 7.28 (1H, d, *J* = 1.1Hz, H-5), 6.75 (1H, d, *J* = 8.3Hz, H-6), 3.39 (2H, m, H-3′), 5.13 (1H, t, *J* = 6.2Hz, H-2′), 1.60 (3H, s, H-4′), 1.64 (3H, s, H-5′), 5.98 (1H, m, H-3′′), 1.89 (3H, d, *J* = 7.32, H-3′′-CH3), 1.67 (3H, s, H-2′′-CH3); ^13^C-NMR (150 MHz, CDCl_3_) *δ*:161.2 (C-2), 113.5 (C-3), 144.1 (C-4), 128.9 (C-5), 106.7 (C-6), 164.0 (C-7), 112.1 (C-8), 151.2 (C-9), 113.5 (C-10), 89.3 (C-2′), 27.6 (C-3′), 82.1 (C-4′), 21.2 (C-4′-CH3), 22.3 (C-4′-CH3), 167.2 (C-1′′), 128.6 (C-2′′), 137.7 (C-3′′), 15.6 (C-2′′-CH3), 20.5 (C-3′′-CH3). Comparing the above data with previous study [24], it was identified as columbianadin.

### 2.7. Anti-Inflammation Activity of the Three Coumarins

The inflammation theory has always attracted widespread attention. Meanwhile, APR plays an important role in the treatment of rheumatoid arthritis and has been reported in many studies [25,26,27]. In order to investigate the anti-inflammation effect of the obtained coumarins from APR extract, CCK-8 assay was first used to assess their cytotoxicity. OE (Figure 7A), CBA (Figure 7B) and CBD (Figure 7C) have no toxic effects within the experimental settings. CBA promoted the proliferation of RAW264.7 cells within a certain range, and the difference was statistically significant (*p* < 0.05), indicating that it promoted the proliferation of macrophages and enhanced their phagocytic function. OE (Figure 7D), CBA (Figure 7E) and CBD (Figure 7F) displayed excellent inhibition activity on NO release at the concentration of 100 μmol/mL (*p* < 0.01), 200 μmol/mL (*p* < 0.01) and 50 μmol/mL (*p* < 0.01), respectively. Meanwhile, the effects of these three coumarins on pro-inflammatory cytokine release in LPS leaded RAW264.7 cell were further investigated. LPS could obviously facilitate the TNF-α, IL-6 and MCP-1 secretion. OE (Figure 7D) and CBA (Figure 7E) also significantly inhibit IL-6 secretion (*p* < 0.05). OE (Figure 7D) could obviously decrease the concentration of TNF-α (*p* < 0.01). OE (Figure 7D), CBA (Figure 7E) and CBD (Figure 7F) could significantly inhibit MCP-1 secretion (*p* < 0.05). These consequences suggest that the three coumarins could potentially be natural anti-inflammatory ingredients that could be used in pharmaceutical products.

## 3. Materials and Methods

### 3.1. Samples and Chemicals

Columbianadin, columbianetin acetate and osthole (puirty ≥ 98% on HPLC) were separated from APR in our lab (Tianjin, China). The angelicae pubescentis radix was purchased from Anguo city (Hebei, China). Ultra-pure water was obtained from a Milli-Q Academic water system (Millipore, Milford, MA, USA). Chromatographic grade acetonitrile, methanol and Analytical grade 95% (*v/v*) ethanol were purchased from Concord Science Co. Ltd. (Tianjin, China). Sodium dodecyl sulfate (SDS) was achieved from Beijing Solarbio science Technology Co. Ltd. (Beijing, China). The RAW264.7 cell was purchased from Shanghai Institute of Cell Biology (Shanghai, China). Lipopolysaccharide (LPS) was purchased from Sigma (St. Louis, MO, USA).

### 3.2. Adsorbents

Three MARs, including D101, DA-201 and AB-8 were obtained from Donghong Chemical Co. Ltd. (Xuzhou, Jiangsu, China). Other two MARs (HP-20 and GDX-201) were purchased from Welch Materials Inc. (Shanghai, China). These MARs were completely steeped in 95% (*v/v*) ethanol for 1 day and then rinsed with ultra-pure water until ethanol was subsequently eliminated.

### 3.3. Preparation of APR Extract

The APR (1.2 kg) was extracted with 70% (*v/v*) ethanol solution (3 × 6 L, 2 h) using heat reflux extraction. The extracting solution was mixed and concentrated under reduced pressure with rotary evaporator until the ethanol was completely removed. The crude extract of APR was obtained by drying the concentrated solution under constant temperature and vacuum. The crude extract was dissolved with SDS ultrapure water (APRE: SDS = 1:0.75) and configured to various concentrations.

### 3.4. HPLC Analysis of Three Coumarin Compounds

An HPLC 1200 system equipped with a HederaTM ODS-2 chromatographic column (4.6 × 250 mm, 5 µm) was used to determine these three coumarins. The condition of chromatographic analysis was as follows: The mobile phase consisted of acetonitrile (A) and water (B) using the following elution program: 22–22% A, 0–15 min, 22–35% A,15-20 min, 35–35% A, 20–25 min, 35–65% A, 25–30 min, 65–65% A, 30–35 min, 95–95% A, 37–40 min, 22–22% A, 42–45 min. The column temperature, flow rate, injection volume and detection wavelength were considered to be 30 °C, 1 mL/min, 10 µL and 325 nm, respectively. The calibration curves for CBA, OE and CBD were Y = 29.889X − 14.862 (r^2^ = 0.9994), Y = 37.671X − 19.319 (r^2^ = 0.9995) and Y = 27.080X − 13.961 (r^2^ = 0.9995).

### 3.5. Static Adsorption and Desorption Experiments

#### 3.5.1. Optimization of Resin

The optimum MAR was selected by the static adsorption and desorption experiment. Firstly, 25 mL sample solution was added into the activated resins (0.5 g each) in 100 mL conical flasks and shaken (100 rpm) for 6 h at 298 K. Secondly, the resins were washed by ultrapure water (100 mL) after reaching the equilibrium of adsorption. Finally, 95% (*v/v*) ethanol (25 mL) was added for desorption experiment. The remaining conditions were the same as the adsorption experiments.

The concentration of three coumarins in initial sample solution, the equilibrium of adsorption and desorption solution were determined by HPLC 1200. Each experiment was operated three times in parallel.

The capacity and ratio of adsorption and desorption were calculated by the following formulas:(1)$$\mathrm{Absorption}\;\mathrm{capacity}:\;Q_{e}=\frac{\left(C_{0}-C_{e}\right)V_{i}}{W}$$
(2)$$\mathrm{Desorption}\;\mathrm{capacity}:\;Q_{d}=\frac{C_{d}V_{d}}{W}$$
(3)$$\mathrm{Absorption}\;\mathrm{ratio}:\;A=\frac{C_{0}-C_{e}}{C_{0}}\times 100\%$$
(4)$$\mathrm{Desorption}\;\mathrm{ratio}:\;D=\frac{C_{d}V_{d}}{\left(C_{0}-C_{e}\right)V_{i}}\times 100\%$$
where Q_e_ and Q_d_ are the absorption and desorption capacity (mg/g); C_0_, C_e_ and C_d_ stand for the initial, equilibrium and desorption concentrations of these three coumarins in the sample solution, respectively (μg/mL); V_i_ and V_d_ are the volume of the initial and desorption sample solution (mL); W stands for the weight of dry resin (g).

#### 3.5.2. Adsorption Dynamics

Pre-treated HP-20 resin (0.5 g) was mixed with 25 mL of APR extract in a conical flask that were rocked at 100 rpm at 298 K for 6 h. An aliquot of supernatant was collected every 10 min for the first 1 h and every 60 min after the first 1 h. Supernatants at each time points were detected by HPLC 1200. Adsorption dynamics were evaluated by the following three dynamics models.

Pseudo first order model:(5)$$Q_{t}=Q_{e}-Q_{e}e^{-k_{1}t}$$

Pseudo second order model:(6)$$\frac{t}{Q_{t}}=\frac{1}{K_{2}Q_{e}^{2}}+\frac{1}{Q_{e}}t$$

Intra particle diffusion model:(7)$$Q_{t}=K_{\mathrm{ip}}t^{0.5}+I$$
where Q_e_ and Q_t_ stand for the adsorption capacities at equilibrium and at time t (mg/g).

#### 3.5.3. Adsorption Isotherms

Pre-treated HP-20 resin (0.500 g) was put into 25 mL of APR extract at different initial concentrations. The conical flask was rocked at 100 rpm in a rocker (298 K, 308 K and 318 K, respectively) for 6 h. After reaching the adsorption equilibrium, the concentration of three coumarins were determined by HPLC 1200. The adsorption properties of resin were assessed by Langmuir, Freundlich, Dubinin Radushkevich (DR).

Langmuir equations:(8)$$\frac{C_{e}}{Q_{e}}=\frac{C_{e}}{Q_{0}}+\frac{1}{K_{L}Q_{0}}$$

Freundlich equations:(9)$$Q_{e}=K_{F}C_{e}^{1/n}$$

DR equations:(10)$$Q_{e}=Q_{0}e^{-K_{DR}\varepsilon^{2}}$$
(11)$$\varepsilon =RT\ln(1+\frac{1}{c_{e}})$$
(12)$$E={\left(2K_{DR}\right)}^{-\frac{1}{2}}$$
where Q_0_ is the theoretical maximum adsorption capacity (mg/g); R represents the gas constant and is equal to 8.314 J/mol·K; T (K) represents the temperature.

The Langmuir adsorption model was further analyzed by the separation factor (R_L_), and the separation coefficient was calculated by the Formula (13). The experimental concentration range (0 < R_L_ < 1) indicates that the adsorption is a preferential adsorption.
(13)$$R_{L}=\frac{1}{1+K_{L}C_{0}}$$

#### 3.5.4. Adsorption Thermodynamics

To understand the adsorption process from the aspect of energy change, the adsorption thermodynamics parameters that Gibbs energy, Entropy and Enthalpy are obtained using the following formulas:K = MK_L_(14)
(15)$$\Delta G^{0}=-RT\mathrm{lnK}$$
(16)$$\Delta H^{0}=\Delta G^{0}+T\Delta S^{0}$$
where K stands for the equilibrium constant (L/mol), M stands for the molecular weight of coumarins (g/mol).

### 3.6. Dynamics Adsorption and Desorption Experiment

The preprocessed HP-20 resins (5 g) were wet-loaded into the glass column (40 cm × 1 cm, 1 BV = 20 mL) to carry out the dynamics adsorption and desorption experiments. Firstly, 100 mL APR extracts solution (the concentration of CBD, OE and CBA were 109.20 μg/mL, 345.06 μg/mL and 81.84 μg/mL, respectively) was flowed through the resin columns at the speed of 2 BV/h. After the adsorption equilibrium, ultrapure water (5 BV) was used to wash the resin columns to remove impurities such as sugar, pigment, etc. Finally, 5 BV different concentration of ethanol (20%, 40%, 60%, 80%, 100%, *v/v*) were used to desorb the three coumarins at a speed of 2 BV/h and the fractions were further concentrated and dried to calculate the content of the three coumarins.

### 3.7. Enrichment of Three Coumarins by HP-20 Resin

For a lab-scale enrichment of three coumarins, a glass column was wet packed with HP-20 resin (1 BV = 900 mL) to separate the three coumarins fraction from the APR extract. A total of 4.5 L of APR extract solution was loaded onto the top of the HP-20 resin column (L50 cm × 6.0 cm) at a speed of 2 BV/h. After the adsorption equilibrium reached, the deionized water (5 BV) and 60% ethanol (5 BV) were used to remove impurities of APR extract, then 80% ethanol (5 BV) was used to desorb the three coumarins from the HP-20 resin column at a speed of 2 BV/h. The 80% ethanol eluting fraction was concentrated by using a rotary evaporator and then dried under vacuum to obtain the three coumarins fraction. The HP-20 resin was revived with 4.5 L of 95% ethanol, and ultra-pure water at a speed of 2 BV/h.

### 3.8. Separation of Three Coumarins by PHPLC

After enriched by the HP-20 resin, further separation of three coumarins were executed on a Shimadzu Prominence LC-20AP PHPLC (Tokyo, Japan) equipped with Agilent Eclipse XDB-C18 (250 × 21.2 mm, 7 μm, Santa Clara, CA, USA) column. The PHPLC condition for three coumarins was as follows: 70% methanol was used as the mobile phase. The flow rate, detection wavelength and injection volume were 8 mL/min, 325 nm and 1.0 mL, respectively.

Finally, the three coumarins structures were identified by confrontation of the spectral characteristics (MS, ^1^H-NMR, and ^13^C-NMR) with those depicted in a previous study. The purities of the three coumarins were determined by HPLC 1200.

### 3.9. The Anti-Inflammatory Effect of the Three Coumarins on RAW264.7 Macrophages

#### 3.9.1. RAW264.7 Cell Viability Assay

Viability was determined by Cell Counting Kit-8 (CCK8) method. Cells (3 × 10^5^ cells/mL) were seeded into 96-well plates (Corning, New York, NY, USA) with 100 µL added per well; the plates were hatched at 37 °C in the presence of 5% CO_2_ for 24 h. Cells were then supplemented with 10 µL CCK8 reagent in each well, then placed in an incubator with 5% CO_2_ at 37 °C for 0.5 h. Absorbance values were measured at 450 nm using a microplate reader (F039003, Tecan, Australia).

#### 3.9.2. Effects of These Three Coumarins on the Release of Inflammatory Factors by LPS Induced in RAW264.7 Cells

The cell suspension was added to a 96-well plate at a cell density of 3 × 10^5^ cells/mL, 100 μL per well. After being cultured in a cell incubator at 37 °C and 5% CO_2_ for 24 h, and dealt with different concentrations of CBD, OE and CBA for 2 h, followed by treatment with LPS and incubated for another 24 h. Then we collected 50 μL of the supernatant of each well and added them to a new 96-well plate. According to the instructions of the Griess assay, Griess Reagent I (50 μL) and Griess Reagent II (50 μL) were put into each well, respectively. And the absorbance was measured at 540 nm with a microplate reader. In addition, the three inflammatory factors (TNF-α, IL-6, MCP-1) were also tested.

### 3.10. Statistical Analysis

Data were expressed as mean ± SD values. Statistical analysis was performed by using GraphPad Prism 5.0 software (GraphPad, San Diego, CA, USA). Significance of the differences between variables was tested by one-way ANOVA. *p* values less than 0.05 was considered statistically significant.

## 4. Conclusions

A simple and useful method was established for simultaneous enrichment and separation of CBA, OE and CBD from APR extract using MAR coupled with PHPLC. Among the five MARs investigated, HP-20 resin showed the best adsorption and desorption capacity for three coumarins. The adsorption dynamics data of CBA, OE and CBD well fitted to the pseudo second order equation. The Langmuir, Freundlich and DR isotherms models could be used to understand the adsorption process of the three coumarins. After enrichment by the HP-20 resin, the content of CBA, OE and CBD increased from 0.27%, 1.15% and 0.36% in the crude APR extract to 2.92%, 22.98% and 7.16% in the 80% ethanol fraction with a recovery yield of 31.04%, 58.05% and 57.17%, respectively. Subsequently, the PHPLC was used to purify the three coumarins from the 80% ethanol fraction treated after the HP-20 resin. The purity of three coumarins could reach above 98%. CBA, OE and CBD could obviously restrain NO release and MCP-1 secretion in LPS leaded RAW 264.7 cells. These three coumarins could be used as potentially natural anti-inflammatory ingredients in pharmaceutical products. The established method was fit for enlarging scale separation and enrichment of these three coumarins with anti-inflammatory effect from APR extract. Furthermore, the technique of combining two methods may be widely used in the separation of other coumarins in natural products.

## Figures and Tables

**Figure 1 molecules-24-02664-f001:**
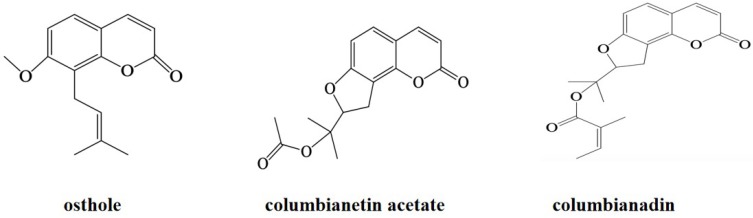
Chemical Structures of three coumarins.

**Figure 2 molecules-24-02664-f002:**
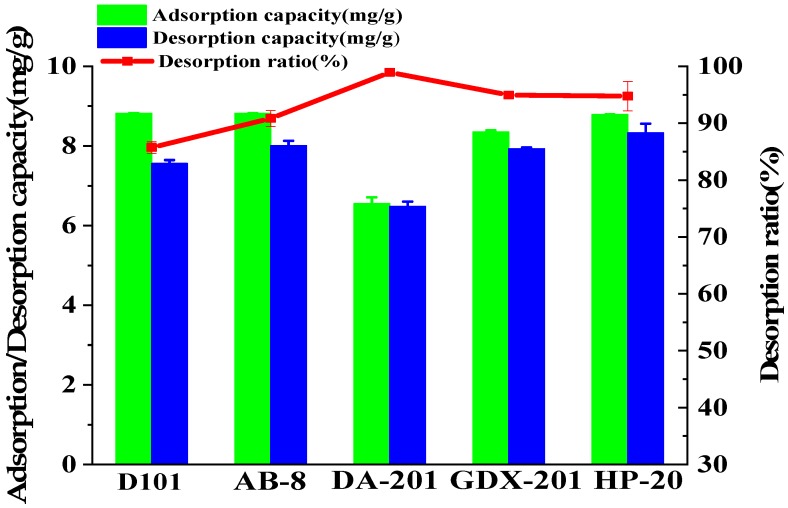
Adsorption, desorption capacities and desorption ratio of the total of three coumarins.

**Figure 3 molecules-24-02664-f003:**
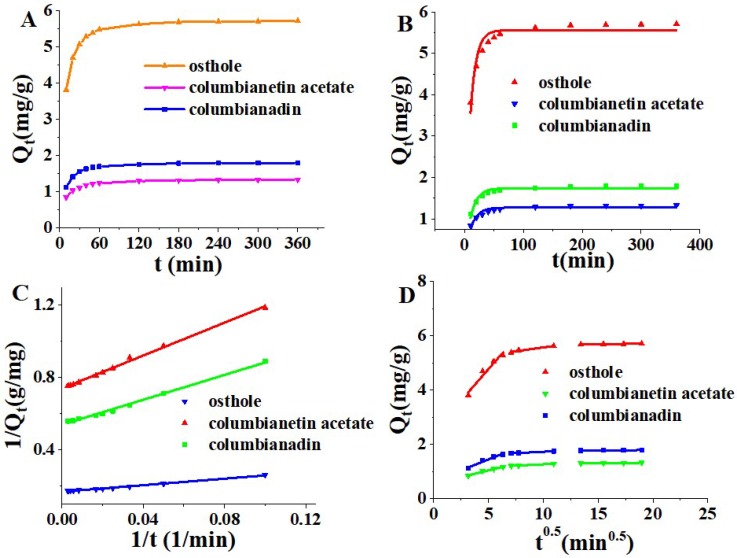
(**A**) Adsorption kinetics curve, (**B**) Pseudo-first-order model, (**C**) Pseudo-second-order model, (**D**) Intra-particle diffusion model of three coumarins on HP-20 resin at 298 K. Adsorption capacity at time t (Q_t_).

**Figure 4 molecules-24-02664-f004:**
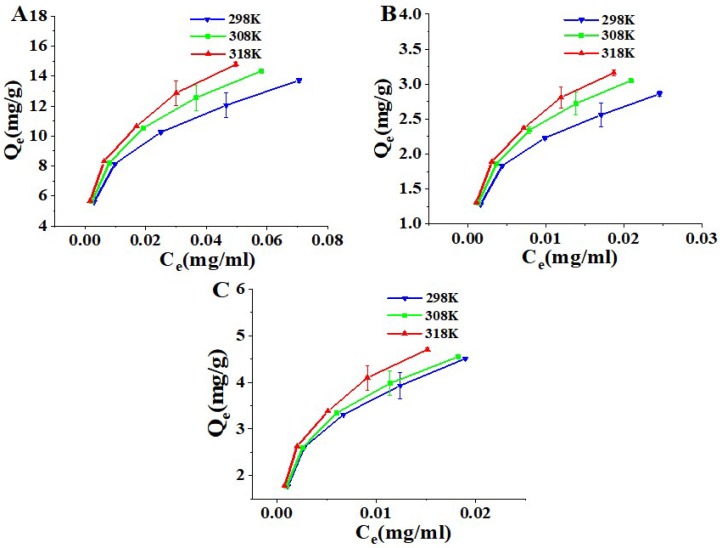
Adsorption isotherms of osthole (**A**), columbianetin acetate (**B**), columbianadin (**C**) on HP-20 resin at different temperature. The equilibrium concentration (C_e_). The adsorption capacities at equilibrium (Q_e_).

**Figure 5 molecules-24-02664-f005:**
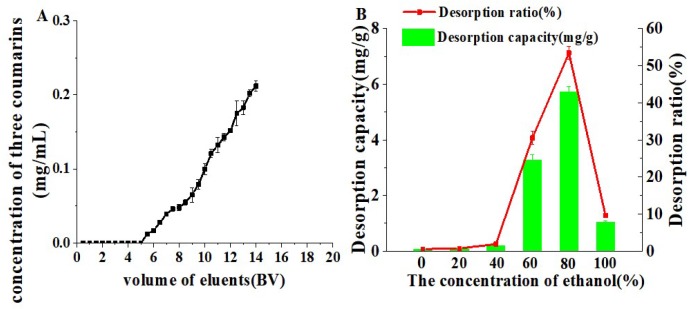
Dynamic breakthrough curve (**A**), the effect of ethanol concentration (**B**).

**Figure 6 molecules-24-02664-f006:**
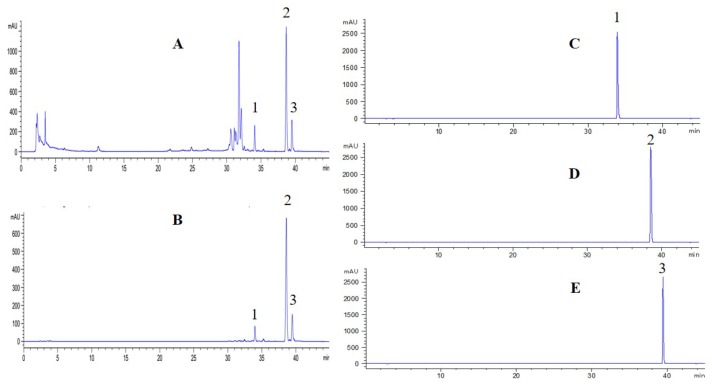
HPLC Chromatograms of samples: Crude extract (**A**); After treatment by HP-20 resin (**B**); Separation by PHPLC (**C**–**E**); 1, 2 and 3 represent for columbianetin acetate, osthole and columbianadin, respectively.

**Figure 7 molecules-24-02664-f007:**
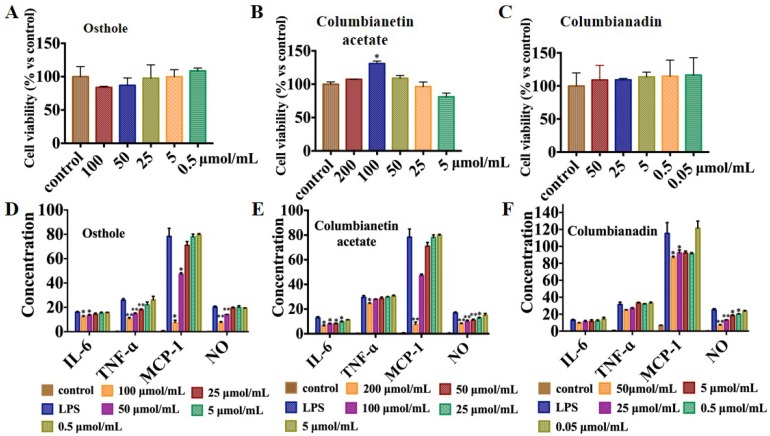
The cell viability of three coumarins (**A**–**C**); The anti-inflammatory effects of three coumarins on IL-6 (ng/mL), TNF-α (ng/mL), MCP-1 (ng/mL) and NO (μM) levels in Lipopolysaccharide (LPS)-stimulated RAW264.7 macrophages (**D**–**F**). Data are presented as mean ± SD of three independent experiments performed in duplicate. * *p* < 0.05 and ** *p* < 0.01 versus LPS group.

**Table 1 molecules-24-02664-t001:** Physical properties of five macroporous resins.

Resins	Structure	Polarity	Particle Size (mm)	Pore Diameter (nm)	Surface Area (m^2^/g)
D101	polystyrene	Non-polar	0.30–1.25	9–15	480–550
AB-8	polystyrene	Weak-polar	0.30–1.25	13–14	480–520
DA-201	polystyrene	Polar	0.30–1.25	10–13	≧200
HP-20	polystyrene	Non-polar	0.30–1.25	29–30	550–600
GDX-201	polydivinylbenzene	Non-polar	0.30–1.25	—	510

**Table 2 molecules-24-02664-t002:** Adsorption kinetics parameters of three coumarins on HP-20 resin at 298 K.

Kinetics Model	Parameters	Columbianetin Acetate	Osthole	Columbianadin
Pseudo-first-order	Q_e_ (mg·g^−1^)	1.2748	5.5582	1.7411
	K_1_ (min^−1^)	0.0880	0.1022	0.0890
	R^2^	0.8424	0.9066	0.9176
Pseudo-second-order	Q_e_ (mg·g^−1^)	1.3508	5.8824	1.8512
	K_2_ (g·mg^−1^·min^−1^)	0.1210	0.0322	0.0851
	R^2^	0.9971	0.9927	0.9942
Intra-particle diffusion (10–360 min)	I (mg·g^−1^)	0.9619	4.4638	1.3361
	K_i_ (mg·g^−1^·min^−0.5^)	0.0233	0.0813	0.0295
	R^2^	0.6952	0.6104	0.5843
Intra-particle diffusion (10–40 min)	I (mg·g^−1^)	0.5348	2.4562	0.6456
	K_i_ (mg·g^−1^·min^−0.5^)	0.1031	0.4633	0.1606
	R^2^	0.9775	0.9538	0.9700
Intra-particle diffusion (50–120 min)	I (mg·g^−1^)	1.0605	5.0046	1.5383
	K_i_ (mg·g^−1^·min^−0.5^)	0.0213	0.0563	0.0192
	R^2^	0.9885	0.9465	0.9796
Intra-particle diffusion (180–360 min)	I (mg·g^−1^)	1.2795	5.6076	1.7495
	K_i_ (mg·g^−1^·min^−0.5^)	0.0026	0.0055	0.0022
	R^2^	0.9580	0.9402	0.8516

Note: The adsorption capacity at equilibrium (Q_e_), the constants for pseudo-first-order, pseudo-second-order and intra-particle diffusion models (K_1_, K_2_ and K_i_ respectively), the boundary layer thickness (I).

**Table 3 molecules-24-02664-t003:** Langmuir, Freundlich and Dubinin Radushkevich (DR) parameters for three coumarins on HP-20 resin at different temperature.

Compound	Temperature (K)	Langmuir Equation				Freundlich Equation			DR Equation			
		R^2^	K_L_	Q_0_ (mg/g)	R_L_	R^2^	K_F_	1/n	Q_0_ (mg/g)	R^2^	K_DR_	E
Columbianetin acetate	298	0.9929	318.90	3.14	0.103	0.9927	8.09	0.2804	4.20	0.9946	4.69 × 10^−9^	10.33 × 10^3^
	308	0.9936	352.14	3.38	0.094	0.9962	9.73	0.2977	4.73	0.9987	4.51 × 10^−9^	10.53 × 10^3^
	318	0.9917	391.10	3.50	0.086	0.9968	10.58	0.3016	4.98	0.9954	4.16 × 10^−9^	10.97 × 10^3^
Osthole	298	0.9898	125.56	14.75	0.065	0.9975	28.29	0.2748	17.14	0.9847	5.62 × 10^−9^	9.43 × 10^3^
	308	0.9901	149.77	15.53	0.055	0.9997	32.27	0.2848	18.59	0.9875	5.19 × 10^−9^	9.81 × 10^3^
	318	0.9872	179.43	15.92	0.046	0.9983	34.47	0.2830	19.27	0.9790	4.58 × 10^−9^	10.45 × 10^3^
Columbianadin	298	0.9911	386.73	4.97	0.066	0.9923	15.17	0.3057	7.09	0.9939	4.81 × 10^−9^	10.20 × 10^3^
	308	0.9933	415.00	5.02	0.062	0.9937	15.56	0.3047	7.23	0.9987	4.45 × 10^−9^	10.60 × 10^3^
	318	0.9911	492.82	5.20	0.053	0.9948	17.42	0.3103	7.71	0.9960	4.08 × 10^−9^	11.08 × 10^3^

Note: Theoretical maximum adsorption capacity (Q_0_), a constant related to the free energy of adsorption (K_L_). the Freundlich constant indicating adsorption capacity (K_F_), an empirical constant demonstrating adsorption intensity of the system (1/n), the DR constant related to the adsorption energy (K_DR_), separation factor (R_L_), adsorption energy (E).

**Table 4 molecules-24-02664-t004:** Thermodynamics Equation parameters for three coumarins on HP-20 resin at different temperature.

Compound	Temperature (K)	∆G^0^ (kJ/mol)	∆H^0^ (kJ/mol)	∆S^0^ (J/mol)
Columbianetin acetate	298	−29.27	8.03	121.97
	308	−29.52		
	318	−29.79		
Osthole	298	−25.60	14.06	133.05
	308	−26.03		
	318	−26.48		
Columbianadin	298	−31.07	9.50	129.45
	308	−31.26		
	318	−31.71		

Note: Gibbs energy (∆G^0^), enthalpy (∆H^0^), entropy (∆S^0^).

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
