# Peer review of "Separation and Enrichment of Three Coumarins from Angelicae Pubescentis Radix by Macroporous Resin with Preparative HPLC and Evaluation of Their Anti-Inflammatory Activity"

_molecules, 2019, doi:10.3390/molecules24142664_

Round 1

Reviewer 1 Report

The manuscript entitled „Separation and enrichment of columbianetin acetate, osthole and columbianadin with anti-inflammatory effect from Angelicae Pubescentis Radix by macroporous resin with preparative HPLC“ refers to the scientific field covered by Molecules journal.

The authors have produced a large body of work and follow a logical sequence to discuss their findings.

The title is to long and confusing so I suggest its change (for example delete „with antiiflammatory effect“ or add it at the end, change compounds names with „coumarins“…). For example „Separation and enrichment of three coumarins from Angelicae Pubescentis Radix by macroporous resin with preparative HPLC and their anti-inflammatory activity/effect“.

There are lot of typos in the manuscript so it should be checked ones more.

Use the full word (name) only the first time it appears, and further you should use abbreviations.

Full names of all the goods, providers and manufactures (of instruments (columns), softwares, reagents and chemicals) should be included (Model, Company, Town, State).

Tables and Figures should be fully understandable without reading the text. Please avoid abbreviations and/or write footnotes with explanations under the tables and figures.

Line 106 – why the authors used Kelvins when in table 2 and 4 are °C. Please correct.

It is „mol“ and not „moL“ (capital letter L), it is „kJ“ and not „KJ“, etc.

Some particular points of future research could also be added in the conclusion section.

Reviewer 2 Report

In this work, the authors reported a promising method for separation and enrichment of columbianetin acetate, osthole and columbianadin from APR extract by combining macroporous resins with preparative HPLC. The anti-inflammatory activity of the three coumarins was further evaluated by LPS-induced RAW 264.7 cells, suggesting that the three coumarins could be used as potentially natural anti-inflammatory ingredients. This research is interesting. I recommend it for acceptance after minor revision as detailed below.

-Page 9, the section of 3.7 Anti-inflammation activity of the three coumarins. It is suggested to describe the results of the three coumarins in the order as shown in Fig. 7, i.e. OE (A), CBA (B) and CBD (C). Also, “CBA (Fig.7E) and OE (Fig.7D) also 207 significantly inhibit IL-6 secretion. CBA, OE and CBD could significantly inhibit MCP-1 secretion”. p-value should be provided to show the significance.

-The resolution of Figs 3 &4 is low. Figures are required at high resolution.

-In Figs 4-6, the panels of each figure should be clearly indicated using labels A, B, C, etc, instead of starting with G, H, or J.
